# Transient chemical and structural changes in graphene oxide during ripening

Hayato Otsuka [1], Koki Urita [2], Nobutaka Honma[3], Takashi Kimuro[4], Yasushi Amako [5], Radovan Kukobat [1,6], Teresa J. Bandosz [7], Junzo Ukai[3], Isamu Moriguchi [2] & Katsumi Kaneko [1] ✉

Graphene oxide (GO)—the oxidized form of graphene—is actively studied in various fields, such as energy, electronic devices, separation of water, materials engineering, and medical technologies, owing to its fascinating physicochemical properties. One major drawback of GO is its instability, which leads to the difficulties in product management. A physicochemical understanding of the ever-changing nature of GO can remove the barrier for its growing applications. Here, we evidencde the presence of intrinsic, metastable and transient GO states upon ripening. The three GO states are identified using a $\pi - \pi^*$ transition peak of ultraviolet–visible absorption spectra and exhibit inherent magnetic and electrical properties. The presence of three states of GO is supported by the compositional changes of oxygen functional groups detected via X-ray photoelectron spectroscopy and structural information from X-ray diffraction analysis and transmission electron microscopy. Although intrinsic GO having a $\pi - \pi^*$ transition at 230.5 ± 0.5 nm is stable only for 5 days at 298 K, the intrinsic state can be stabilized by either storing GO dispersions below 255 K or by adding ammonium peroxydisulfate.

Graphene oxide (GO) is a fascinating material with myriad applications in electronic devices[1,2], supercapacitors[3,4], desalination membranes[5,6], composite materials[6–10], electrocatalysts[11], catalyst supports[12], gas sensors[13] and a broad range of medical therapies[14]. Although Brodie first described the synthesis of GO, Hummers is credited with popularizing research into GO, especially during the last two decades[15,16].

GO has a unique single atomic layer structure with various oxygen-containing functional groups (OFGs), including epoxide, hydroxyl, and carboxyl groups[17,18] and exhibits excellent interfacial properties[19–23]. Despite numerous studies of the surface chemistry of GO[17,18], its exact chemical structure is still not completely understood with its unstable nature playing an important role[24]. Although its functional groups render GO hydrophilic compared to other graphitic materials, providing scope for a broad range of modifications and applications[1–14], they are also responsible for its instability, which typically causes the colour of GO colloidal suspensions to change from light brown to black as a result of spontaneous reactions that occur even under ambient conditions[25–31]. Exposure to light/irradiation promotes photoreactions between the OFGs and conjugated carbon frames[25], as well as chemical reactions with water molecules or hydroxyl ions[27,32], which are enhanced by increasing the pH[27] and temperature[28] of the suspension. Kim et al. synthesized multilayer GO by the oxidation of graphene films grown epitaxially on the C-terminated surface of a Si-C wafer[29]. This multilayer GO was a

[1]Research Initiative for Supra-Materials, Shinshu University, 4-17-1 Wakasato, Nagano, Nagano 380-8553, Japan. [2]Graduate School of Engineering, Nagasaki University, 1-14 Bunkyo-machi, Nagasaki, Nagasaki 852-8521, Japan. [3]New Material & Value Creation Gr., Project Material Creation Dept., Mobility Material Engineering Div., Toyota Motor Corporation, 1, Toyota-cho, Toyota, Aichi 471-8572, Japan. [4]Development Gr.2, Development Section, Engineering Dept., Sanwayuka Industry Corporation, Fukada 15, Ichiriyamacho, Kariya, Aichi 448-0002, Japan. [5]Department of Physics, Faculty of Science, Shinshu University, 3-1-1 Asahi, Matsumoto, Nagano 390-8621, Japan. [6]Department of Chemical Engineering and Technology, Faculty of Technology, University of Banja Luka, B.V. Stepe Stepanovica 73, 78 000 Banja Luka, Bosnia and Herzegovina. [7]Department of Chemistry and Biochemistry, The City College of New York, 160 Convent Avenue, New York, NY 10031, USA. ✉e-mail: kkaneko@shinshu-u.ac.jp

metastable material whose structure and chemistry evolved at room temperature with a characteristic relaxation time of approximately one month. Despite its importance, investigations of the instability of GO colloids are rather scarce, with only a few studies having directly examined the chemical and structural metastability of GO upon ripening, either under ambient conditions[29,30,32–35] or slightly elevated temperatures[28,31]. Upon aging, epoxy groups are converted into –OH (with the help of matrix hydrogen)[29,33,35] and decomposed/desorbed from the lattice, decreasing the interlayer distance and the defect level[35]. Chemical changes are caused by the diffusion of oxygen[33,35] and the possible migration of C–O bonds[32] with a negligible negative effect on the lattice size[32]. Moreover, diffusion causes phase transformation, reflected by the existence of distinct oxidized and graphitic domains in the aged material[28]. The extent of the changes at room temperature is affected by the storage conditions, including air and light exposure[33]. The exposure of reduced graphene oxide (rGO) to water for 90 days is known to result in an increased number of oxygen groups and decreased conductivity[34]. Because colloidal changes proceed gradually with time and alter the properties of GO-containing composite devices[30,31], the lifetime of GO colloids after their synthesis should be thoroughly investigated to produce more stable colloidal GO.

In this study, we investigated the ripening of GO to identify its distinct chemical and structural stages and advanced the understanding of and control over the instability of GO. The spontaneous changes in GO colloids ripened at different temperatures were investigated based on the $\pi - \pi^*$ transition of GO using ultraviolet–visible (UV-Vis) spectroscopy[36,37]. Detailed analyses of the UV-Vis spectra revealed the presence of three distinctive states: An unstable intrinsic state, a metastable state before starting reduction and a transient state that transforms into rGO. Prolonging the lifetime of GO colloids is also explored and an effective method for the suppression of spontaneous changes is proposed.

## Results

### Changes in the optical properties upon ripening

The as-prepared GO colloids are yellow-brown and their color does not clearly change before 1 h of ripening, even at elevated temperatures (Fig. 1a). The color turns from yellow brown to light brown after ripening for 3 h at 333 K and 348 K; the GO ripened at 348 K for 2 days becomes black, whereas that ripened at 298 K remains brown. A slight color change in the GO ripened at 298 K is observed after 7 days, whereupon the color of GO ripened at 333 K and 348 K is black.

Changes in the UV-Vis absorption spectra of the GO colloids ripened for different durations at 298 and 348 K are investigated (Fig. 1b, c). The spectral changes of all GO colloids ripened under different conditions are shown in Supplementary Fig. S1. Unripened GO colloids (0 h) show the main absorption peak at 230.4 ± 0.2 nm assigned to the $\pi - \pi^*$ transition of the aromatic C–C bonds; the broad shoulder observed around 300 nm is attributed to the $n - \pi^*$ transition of the C=O bonds. A slight change is observed in the absorption spectra of the GO colloids ripened for different durations at 298 K (Fig. 1b and Supplementary Fig. 1a) and 308 K (Supplementary Fig. 1b). The intensity of the broad shoulder in the visible region increases with ripening time at 333 K (Supplementary Fig. 1c) and 348 K (Fig. 1c and Supplementary Fig. 1d), resulting in the broad absorption tail up to 600 nm due to the dark color of GO[28]. Magnified spectra in the rage of 200–260 nm and 280–300 nm are shown in Supplementary Fig. 1e–h, i–l, respectively.

Systematic analysis of the UV-Vis absorption spectra of the ripened GO colloids provides insights into the instability of GO colloids. The UV-Vis spectra are deconvoluted based on the peak positions at 230 nm and 300 nm as the changes in these peaks reflect the structural changes in the GO framework upon ripening. An example of the deconvoluted UV-Vis spectrum of the GO colloids ripened at 348 K for 48 h is shown in Supplementary Fig. 2. Figure 2 shows the peak positions of the $\pi - \pi^*$ (Fig. 2a) and $n - \pi^*$ transitions (Fig. 2b) as a function of the ripening time after peak deconvolution. These peak

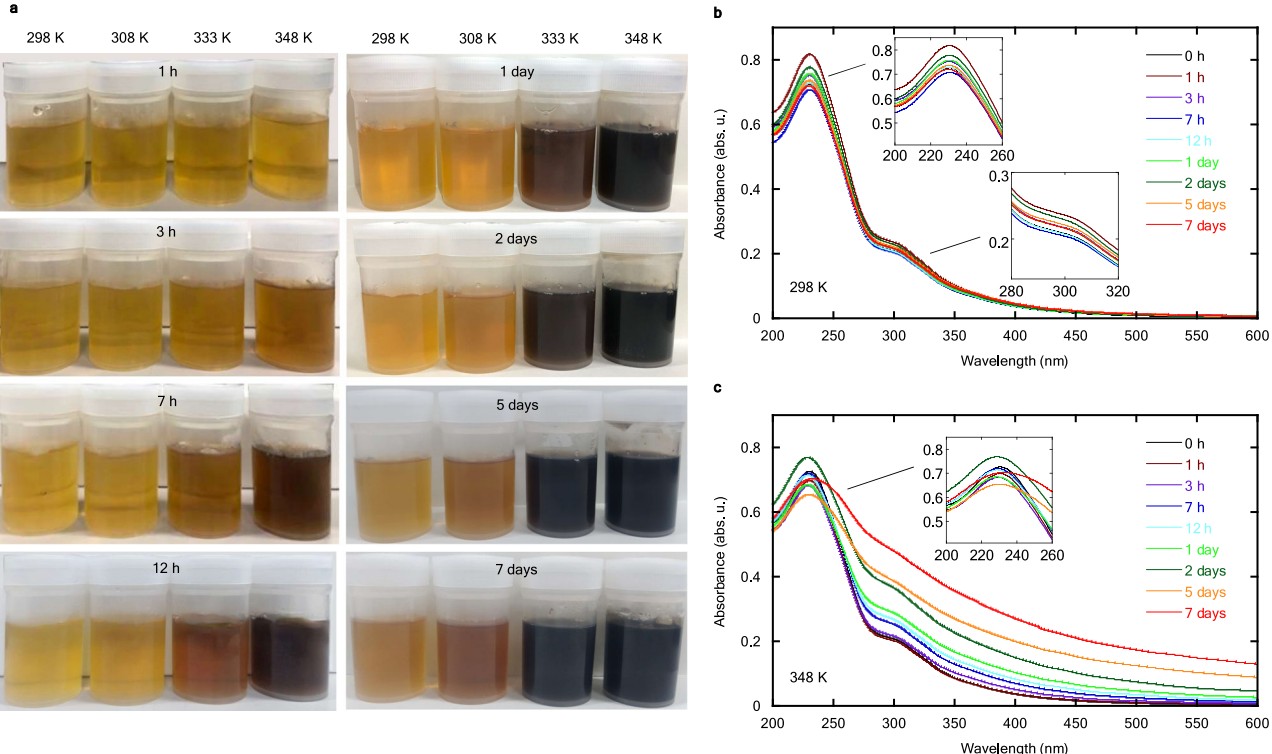

**Fig. 1 | Changes in the optical properties of GO colloids. a** Photographs of the GO colloids ripened at different temperatures and for different durations. **b, c** UV-Vis absorption spectra of GO colloids ripened at 298 K (**b**) and 348 K (**c**) for different durations. The ripening time: black, 0 h; brown, 1 h; purple, 3 h; blue, 7 h; pale blue, 12 h; yellow-green, 1 day; green, 2 days; orange, 5 days and red, 7 days.

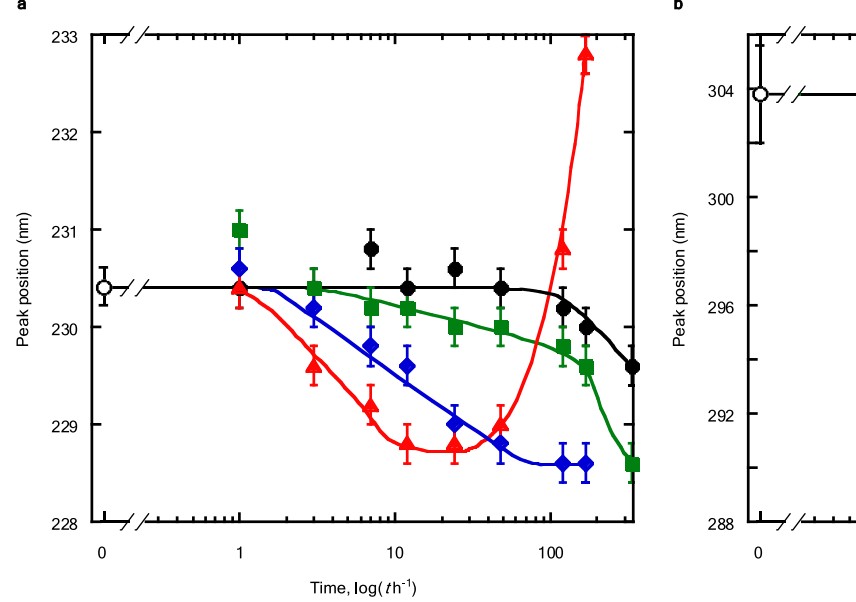
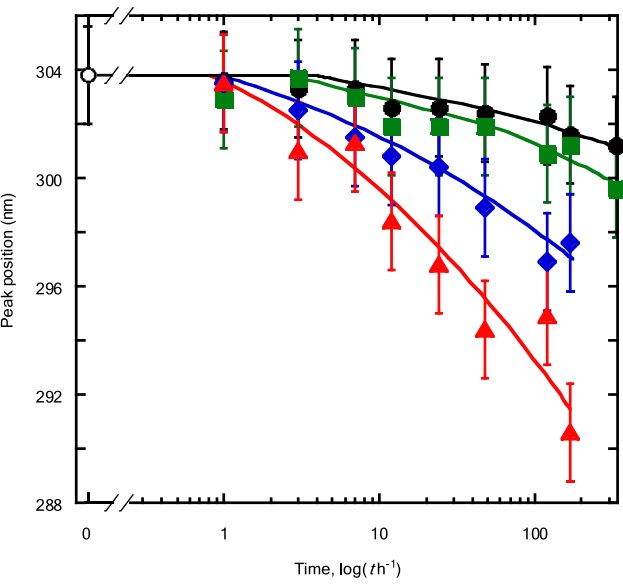

**Fig. 2 | Changes in the peak position with ripening time. a** $\pi - \pi^*$ transition. **b** $n - \pi^*$ transition. Horizontal axis is a logarithmic scale. Ripening temperature: black solid line and circles, 298 K; green solid line and squares, 308 K; blue solid line and diamonds, 333 K; red solid line and triangles, 348 K. Open circle represents the peak position of the GO colloid before ripening. Error bars represent the reproducibility in the peak position determination of at least three measurements.

positions change with the ripening time and temperature. The peak positions of the $\pi - \pi^*$ transition of the GO ripened at 298 K are essentially constant for 100 h, whereas that of the GO ripened at 308 K and 333 K gradually blue-shifts after 7 h and 3 h, respectively. The peak positions of the GO ripened at 348 K reveal an explicit blue-shift after 1 h, and the largest shift is observed at 40 h, whereafter the peak becomes red-shifted. Ripening at 348 K results in a significant change in the $\pi$-conjugated structure, which is associated with the compositional change of OFGs as discussed below. The changes upon ripening at 348 K are discussed with a focus on OFG changes induced by modification of the $\pi$-conjugated structure there.

The change in the $n - \pi^*$ peak position as a function of the ripening time follows a similar trend at all temperatures (Fig. 2b). Higher ripening temperatures (348 K) result in a more pronounced blue-shift, indicating the formation of a stable $\pi$-electronic ground state during ripening.

## Presence of the intrinsic and metastable states of GO

As the peak positions of the $\pi - \pi^*$ transition reflect the ripening-induced structural changes, quantitative analysis of these peak positions can provide information on the ripening process. Figure 3 shows the reduced time course ($t_{red}$) of the peak position of the $\pi - \pi^*$ transition of GO colloids. The reduced time courses are obtained by compressing the time course using the ripening time when the peak position reaches 229.6 nm (except for ripening at 348 K), which corresponds to the intermediate energy between the initial and lowest peak positions. GO is assumed to vary following the same time track, regardless of ripening temperature and a reduced time is then introduced using a definite energy level. The ripening times at the 229.6 nm peak position for 298, 308, 333 and 348 K are 336, 168, 12 and 3 h, respectively and the time compression ratios are 1/122 (298 K), 1/56 (308 K) and 1/4 (333 K); see Supplementary Note 1.

All peak positions at different ripening temperatures are well expressed by the single reduced time course. The peak position vs. the reduced time course indicates the presence of three GO states: The initial stable region up to 1 h, the blue-shift region from 1 h to approximately 40 h and an upward shift after 40 h. The GO corresponding to three states should have individual $\pi$-conjugated

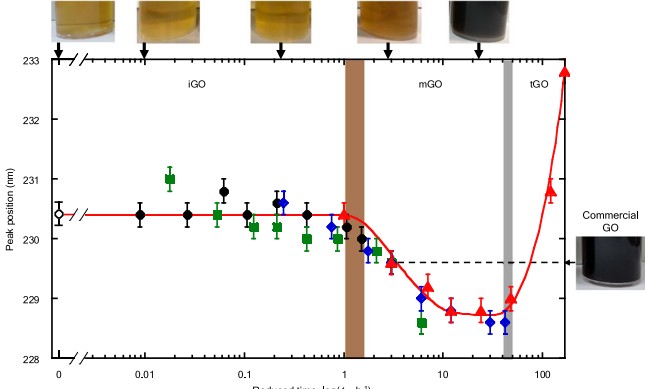

**Fig. 3 | Reduced time course of the peak position of the $\pi$-$\pi^*$ transition of GO colloids ripened at different temperatures for various durations.** The horizontal axis is the logarithm of the reduced time ($t_{red}$). The reduced time course consists of three regions attributed to intrinsic GO (iGO), metastable GO (mGO) and transient GO (tGO). Brown and gray zones represent the transition regions. Photographs of the GO colloids are shown on top of the graph; the arrows indicate the corresponding reduced time. A photo of commercial GO is shown on the right side, and the peak position of its $\pi - \pi^*$ transition is indicated by an arrow and a dashed line. Ripening temperature: black circles, 298 K; green squares, 308 K; blue diamonds, 333 K; red triangles, 348 K. Open circle represents a peak position of the non-ripened GO colloid. Error bars represent the reproducibility in the peak position determination of at least three measurements.

electronic structures, as the $\pi - \pi^*$ transition is assigned to the electronic transition of the conjugated frame structure of GO.

It is important to quantitatively show the stability period of the GO grown at an early stage referred to as intrinsic GO (iGO) and the following blue-shift region as metastable GO (mGO)[28,29]. The lifetimes of iGO and mGO at different ripening temperatures are estimated (Supplementary Table 1). The preservation time of the mGO colloids at 298 K (190 days) is longer than the relaxation time (1 month) of the reported metastable GO film prepared on a SiC wafer[29]. This difference was attributed to the ripening condition as the GO film was aged under dry

conditions and not in an aqueous solution. The red-shift region, which is only observed in the GO colloids ripened at 348 K for longer than 40 h, is attributed to transient GO (tGO) as the GO colloids transforming into the reduced form of GO. We compared the properties of iGO, mGO and tGO with those of commercial GOs. Commercial GOs exhibit a $\pi - \pi^*$ transition peak at $229.6 \pm 0.2$ nm (Supplementary Fig. 3b), corresponding either to mGO or tGO (Fig. 3). Although the commercial GOs appear similar to tGO in both the color and absorption spectra (Supplementary Fig. 3), the commercial GOs are different from tGO, considering various properties given in later (see below).

### Conversion of iGO into mGO with respect to structural change

The changes in the band intensities of the $\pi - \pi^*$ and $n - \pi^*$ transitions with the reduced time course are shown in Fig. 4a, b. The $\pi - \pi^*$ transition intensities gradually decrease in the iGO and mGO regions, and then steeply decrease in the tGO region. The $n - \pi^*$ transition

intensities are essentially constant in the iGO region, increase with the reduced time in the mGO region and reach a maximum value before decreasing in the tGO region. As the change in the $\pi - \pi^*$ transition is associated with the $\pi$-conjugated structure of GO, the three GO states show characteristic changes in the absorption band intensities of the $\pi - \pi^*$ and $n - \pi^*$ transitions, which is related to the partial detachment of the surface functional groups and growth of the graphene-like structure, as mentioned in the Discussion. Supplementary Fig. 4 shows changes in the peak intensity ratio of the $\pi - \pi^*$ and $n - \pi^*$ peaks upon ripening with time and the reduced time. The ratio is constant in the iGO region and decreases in the mGO and tGO regions, indicating the preservation of the $\pi$-conjugated structure in iGO.

The changes in the total oxygen content and the composition of the OFGs determined by X-ray photoelectron spectroscopy (XPS) are shown in Fig. 4c, d, respectively. The C1s and O1s XPS spectra are shown in Supplementary Fig. 5. The total oxygen content is constant in

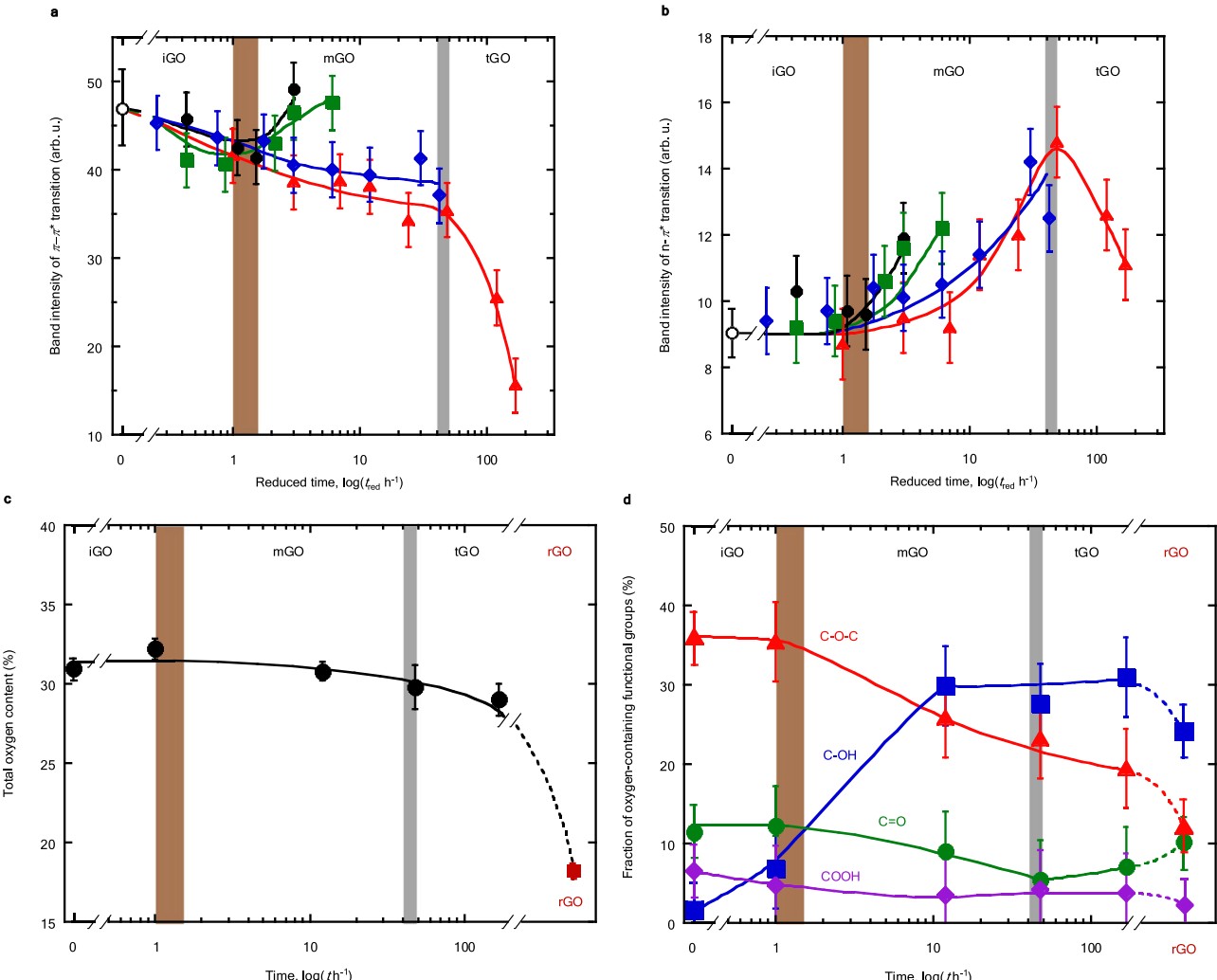

**Fig. 4 | Changes in the structure and OFGs upon ripening. a, b** Band intensity changes of iGO, mGO and tGO with the ripening time. The horizontal axes are expressed as the logarithm of the reduced time ($t_{red}$). **a** $\pi - \pi^*$ transitions. **b** $n - \pi^*$ transitions. Ripening temperature: black solid line and circles, 298 K; green solid line and squares, 308 K; blue solid line and diamonds, 333 K; red solid line and triangles, 348 K. Band intensities of the non-ripened GO colloid are shown with open circle. Error bars in **a, b** represent the reproducibility in the band intensity-determination of at least three measurements. **c** Total oxygen content determined by XPS. Black circles, GO; brown square, rGO. Solid and dashed lines represent the changes upon ripening and reduction, respectively. **d** Compositional changes of

the oxygen-containing functional groups determined from the deconvoluted C1s spectra. The fraction of carbon atoms not bonded to oxygen atoms is excluded. Oxygen-containing functional groups: red lines and triangles, epoxy group (C–O–C); blue lines and squares, hydroxyl group (C–OH); green lines and circles, carbonyl group (C=O); purple lines and diamonds, carboxyl group (COOH). Solid and dashed lines represent the changes upon ripening and reduction, respectively. The horizontal axes in **c, d** are expressed as the logarithm of the ripening time of GO ripened at 348 K. The error bars in **c, d** represent the standard deviation of three measurements at different positions. Brown and gray zones represent the transition regions.

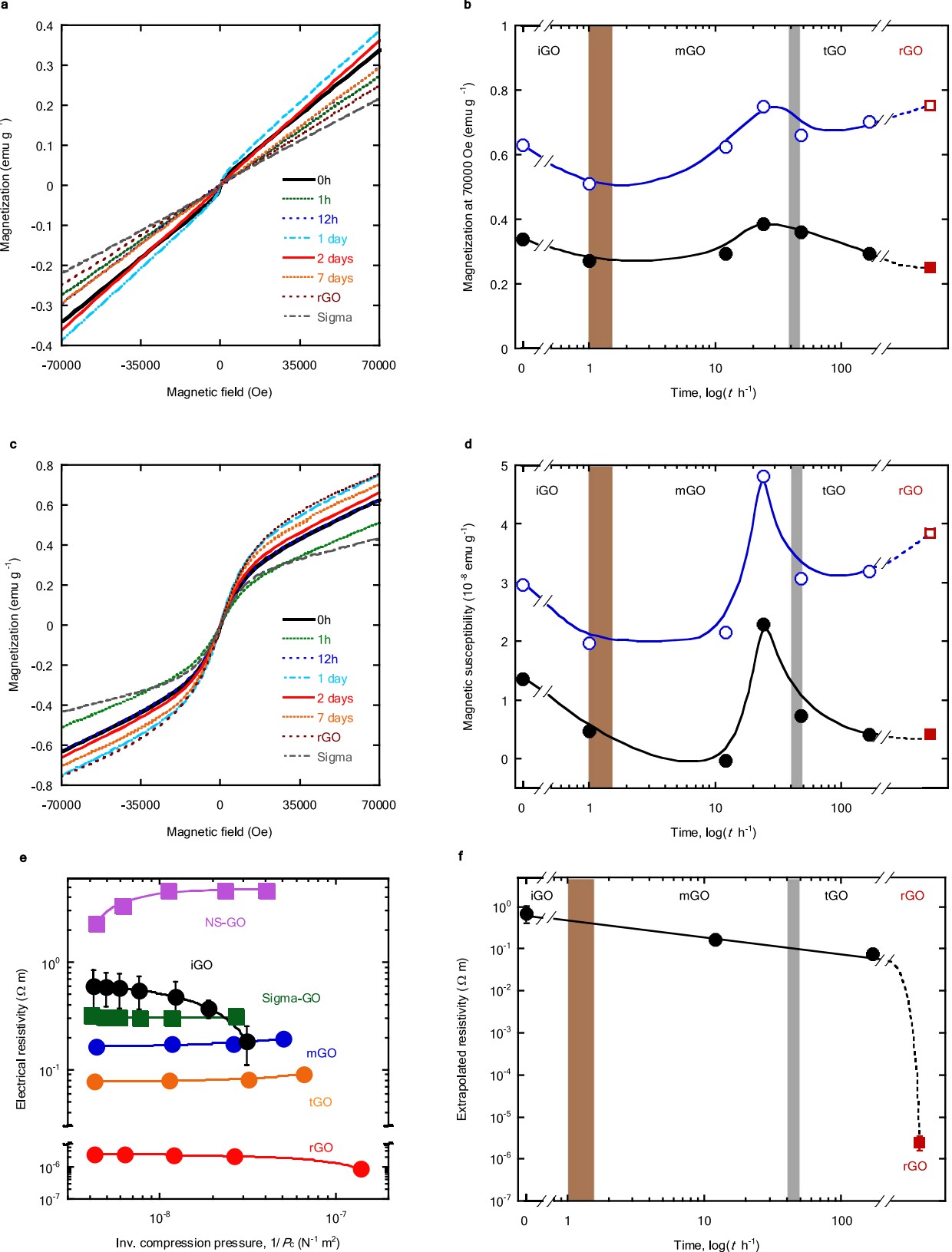

the iGO regions and gradually decreases in the mGO and tGO region. However, epoxy (C–O–C) and hydroxyl (C– OH) groups exhibit different trend; C–O–C decreases with ripening, while C–OH increases[38–40] in the iGO and mGO regions. Conversely, the number of carbonyl and carboxyl groups slightly decrease in the mGO region, although the band intensity of the n − π* transition increases in that

region. Hence, the band intensity of the n − π* transition is not only assigned to the C=O bonds but also to other OFGs.

## Magnetic and electrical properties of GO in the three states

Magnetic properties strongly depend on the electronic structure and are sensitive to the OFGs; each GO state should exhibit different

**Fig. 5 | Magnetic and electrical properties of freeze-dried GO ripened at 348 K and rGO. a, c** $M$-$H$ curves of the freeze-dried GO ripened at 348 K and rGO at 298 K and 1.8 K, respectively. Samples: black solid line, 0 h; green broken line, 1 h; blue dotted line 12 h; pale blue dash-dotted line, 1 day; red solid line, 2 days; orange broken line, 7 days; brown dotted line, rGO; gray dash-dotted line, Sigma-GO. **b** Changes in the magnetization at 70,000 Oe. **d** Changes in the magnetic susceptibility of the GO ripened at 348 K from +500 Oe to −500 Oe. Measurement temperatures: black circles, 298 K; blue open circles, 1.8 K. Magnetizations and magnetic susceptibilities of the rGO at 298 K and 1.8 K are designated a brown square and brown open square, respectively. **e** Electrical resistivities of pelletized GO samples as a function of the reciprocal of compression pressure ($P_c$). The electrical resistivities were measured at compression pressures in the range of 10–240 MPa via the two-electrode method. Black circles, iGO; blue circles, mGO; orange circles, tGO; red circles, rGO; green squares, Sigma-GO; purple squares, NS-GO. **f** Changes in the extrapolated electrical resistivity of the freeze-dried GO ripened at 348 K (black circles) and rGO (brown square). Here, the iGO sample is the non-ripened GO and the mGO and tGO samples are the GOs ripened at 348 K for 12 h and 7 days, respectively. The error bars of iGO and rGO in **e**, **f** represent the reproducibility of duplicate measurements. Brown and gray zones in **b**, **d**, **f** represent the transition regions.

magnetic properties. The magnetic properties were measured after freeze-drying to avoid the effects of aqueous dispersions and ripening during measurement. The $M$-$H$ curves of the GO ripened at 348 K and rGO at 298 and 1.8 K are shown in Fig. 5a, b, respectively, revealing the paramagnetic behavior of GO and rGO. Magnetizations at the maximum magnetic field and magnetic susceptibilities vary with the ripening time. Figure 5c, d show the time courses of the magnetization ($M$) at 70,000 Oe and the magnetic susceptibility ($\chi$) in a magnetic field of +500 Oe to −500 Oe, respectively. Both $M$ and $\chi$ exhibit similar trends over time; the values of iGO decrease with the ripening time, whereas those of mGO reach a maximum and those of tGO gradually decrease. These changes can be attributed to changes in the OFGs, as discussed later.

The electrical properties of GO are closely related to the $\pi$-conjugated electronic structure and the composition of OFGs, which should support the presence of three GO states. The direct-current electrical resistivities of pelletized GO samples were measured using the two-electrode method under various compression pressures. The pelletized GO specimen was prepared from freeze-dried powder of the iGO, mGO, tGO, and commercial GOs. Figure 5e presents the electrical resistivities of GO samples in different states as a function of the reciprocal of the compression pressure. The data of rGO and commercial GOs are also shown, for comparison. The electrical resistivity of iGO increases with the compression pressure, which is different from mGO and tGO. The compression of mGO and tGO improves the interparticle contact and then their electrical resistivities slightly decrease with the compression pressure. The absolute electrical resistivities of iGO, mGO and tGO are much higher than that of the electrically conductive rGO. The reverse tendency of the electrical resistivity of iGO should be associated with the flexible nature[41], however, this point needs to be investigated more quantitatively in the future. Here, we extrapolated the plot of electrical resistivity versus reciprocal compression pressure $P_c$ to the ordinate ($P_c = \infty$) to obtain the minimum contact resistance value, although does not depend on $P_c$. Figure 5f shows the extrapolated resistivity. The resistivity at $P_c = \infty$ gradually decreases with ripening and drops for rGO, indicating a significant difference between the three GOs and rGO.

### Structural differences in the three GO states

Figure 6a shows the changes in the average number of layers and the interlayer distance of GOs and rGO from the X-ray diffraction (XRD) (see Supplementary Fig. 6). The average number of stacking layers was calculated from the stack height ($Lc$) and the interlayer distance. Details are presented in the Supplementary Information (see Supplementary Note 2). The average number of stacking layers decreases monotonically upon ripening, suggesting spontaneous exfoliation of GO in an aqueous media. The initial number of stacking layers of the non-ripened GO is 14, which becomes 6 after ripening for 7 days. All three GO states undergo exfoliation. The interlayer distance is unchanged in the iGO and mGO regions but decreases to 0.74 nm in the tGO region after ripening for 7 days, which is close to that of rGO (0.71 nm).

Figure 6b shows a transmission electron microscopy (TEM) image of non-ripened GO, indicating the presence of a stacking structure of disordered layers. Erickson et al. observed the in-sheet mixed structures of the oxidized and $\pi$-conjugated graphitic regions of GO by TEM[42].

Small changes in the disordered layers are difficult to observe; however, observation of the disordered structure after high-temperature annealing offers a promising route to elucidate these differences (Supplementary Fig. 7) because the high-temperature annealing emphasizes any structural difference in disordered structures, which is known as a heredity effect[43]. The structural changes in GO ripened at 348 K for 2 and 7 days were examined after annealing at 2073 K in Ar. Evident structural differences are observed after ripening (Fig. 6c–e), with marked differences in the number of stacking layers and surface flatness. Ripening for a longer time decreases the number of stacking layer and produces smoother surface. The average number of stacking layers determined from XRD also decreased upon ripening, which is in good agreement with the TEM observation as summarized in Table 1.

Electron energy-loss spectroscopy (EELS) of carbon materials can provide the $sp^2/sp^3$ ratios. The ratio at the position pointed out in Fig. 6f–h is obtained from the areas of $sp^2$ ($\pi^*$) and $sp^3$ ($\sigma^*$) in the EELS spectra shown in Supplementary Fig. 8, following an established procedure[44]. The position number and the measured area are also shown in Supplementary Fig. 8. Figure 6i–k shows the $sp^2/sp^3$ ratios of the edge region of a graphene sheet in the GOs. Figure 6i,j exhibit no significant change in the $sp^2/sp^3$ ratio, but Fig. 6k shows a distinct increase in the ratio towards the edge of graphene sheets. Here, the $sp^2/sp^3$ ratio obtained from the EELS analysis is lower than that obtained from the XPS analysis because the EELS spectra were measured only at the edge part of GO. Despite the minimal differences between the EELS signals obtained from the edge and bulk positions[45], the higher $sp^2/sp^3$ ratio at the edge of the ripened GO at 348 K after 7 days is trustworthy because the other spectra show no differences.

### Suppression routes of iGO-mGO conversion

The intrinsic state of the GO colloids can be preserved under appropriate conditions. The most effective preservation method is storage at $255 \pm 2$ K in the frozen state in the more light-shielded conditions (<<5 lx) (Supplementary Fig. 9), which guarantees the stability of iGO for at least 1 month. The effectiveness of the freezing method was confirmed by the absence of a peak shift in the $\pi - \pi^*$ transition. The addition of an oxidant, ammonium peroxydisulfate ($(NH_4)_2S_2O_8$), also suppressed the conversion of iGO into mGO (Supplementary Fig. 9c). Future work will need to provide a more detailed characterization of iGO stabilized by freezing or addition of $(NH_4)_2S_2O_8$ to broaden the range of applications of iGO.

## Discussion

The structural conversion from iGO to mGO in aqueous media is dominated by the nucleophilic attack with water molecules or the basicity of water, resulting in cleavage of the epoxy ring or $\pi$-conjugated structures[27,28] and increasing the number of −OH groups. Chemical conversion induces changes in the $\pi - \pi^*$ and n $- \pi^*$ transition peaks. Moreover, this conversion explains the observed changes in the magnetic properties. The presence of the epoxy groups in the

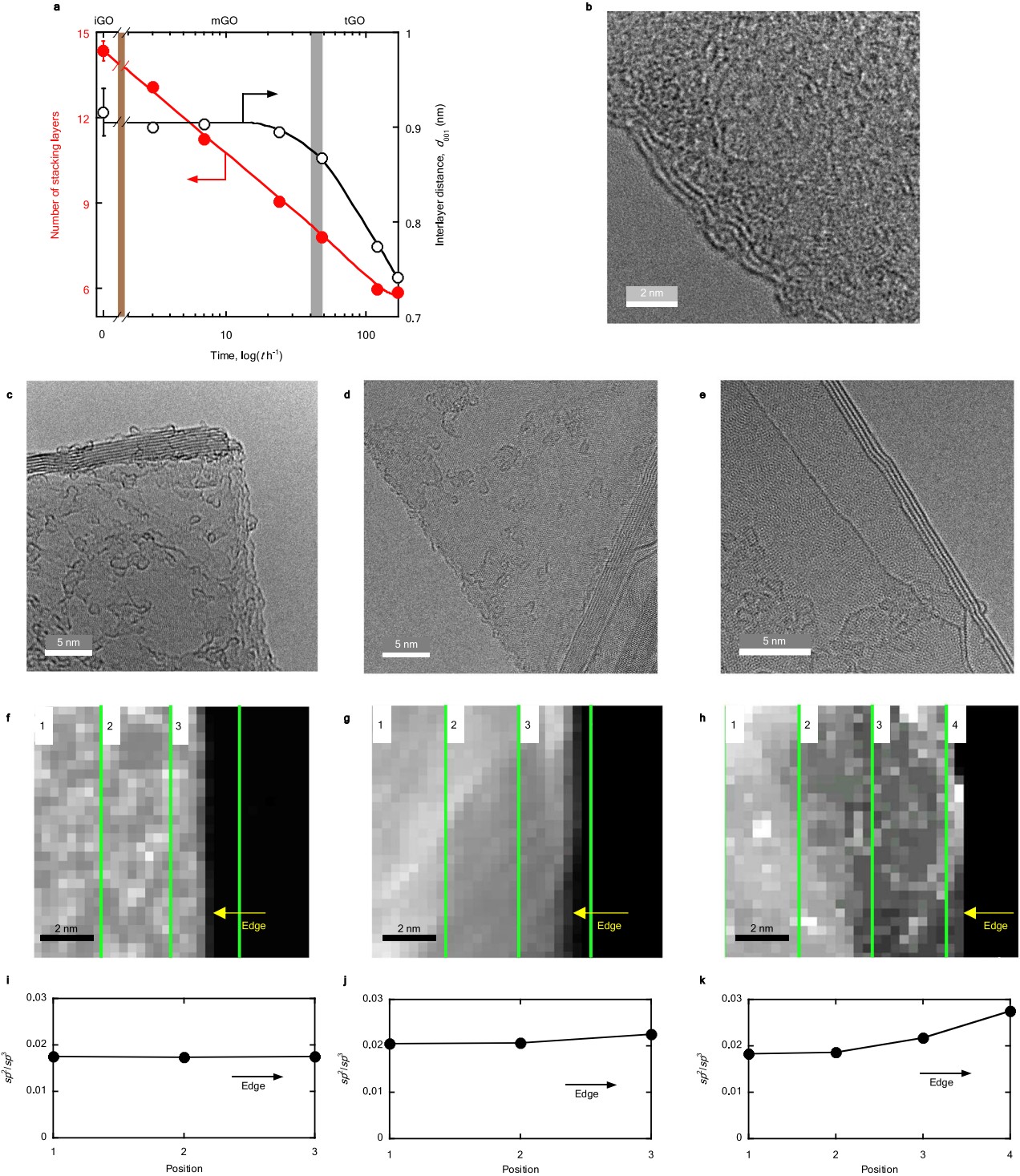

**Fig. 6 | Structural changes upon ripening of GO. a** Number of stacking layers (red circles) and interlayer distances (black open circles) of the GO ripened at 348 K. The error bars of 0 h represent the standard deviation of three measurements. Other error bars are smaller than the symbols. Brown and gray zones represent the transition regions. **b** TEM images of the freeze-dried non-ripened GO powder. **c–e** TEM images of the freeze-dried GO powder annealed in Ar at 2073 K. **f–h** Magnified TEM images containing the edge of annealed GO powder. The observed area is shown with a green-marked region in Supplementary Fig. 8a–c. The images are divided into 4 parts parallel to the edge, and each part is numbered from 1 toward the edge. The boundaries are indicated by green lines. **i–k** The $sp^2/sp^3$ ratio at each position which is indicated in **f–h**. The $sp^2/sp^3$ ratio is obtained from the peak areas of $sp^2$ ($\pi^*$) and $sp^3$ ($\sigma^*$) in the EELS spectra shown in Supplementary Fig. 8. **c, f, i** Non-ripened GO. **d, g, j** GO ripened at 348 K for 2 days. **e, h, k** GO ripened at 348 K for 7 days.

$\pi$-conjugated system causes localization of the $\pi$ electrons and provides carbon radicals due to geometric constraints[46]. The suppression of $M$ and $\chi$ in iGO is attributed to the decomposition of epoxy groups, accompanying the formation of −OH and carbon vacancies. The carbon vacancies produce radicals owing to dangling bonds, resulting in paramagnetic behavior[47,48] with increased $M$ and $\chi$ in the mGO region. The reduction of GO starts in the tGO region, as indicated by a reduction in the oxygen content, which eliminates defects and thereby

decreases the magnetic susceptibility. The three regions correspond to three structurally different GO states, which are completely different from rGO.

Spontaneous exfoliation is caused by changes in the OFGs. Hydrogen bonding of −OH to water molecules is stronger than that of epoxy groups[49] as −OH groups can behave as both an acceptor and a donor of hydrogen. The strength of hydrogen bonding between the GO sheets and water increases with the number of −OH groups. The stabilization of the basal plane of the GO sheets causes exfoliation of the graphene sheets at higher ripening temperatures and durations.

We summarize key properties of iGO, mGO and tGO, which are applicable to their classification in Table 2. Those properties of commercial GOs are shown in the table for comparison. Table 2 shows that various properties are helpful for distinguishing iGO from mGO and tGO. In particular, the peak position of the $\pi - \pi^*$ transition band of the GO colloids is the most reliable characterisitc to identify iGO (see Supplementary Note 3). The information on OFGs from XPS, the magnetic susceptibility, the number of stacking layers based on XRD and TEM and the electrical resistivities of three GO states are mutually different each other. Here, most of the observed values from various methods of Sigma-GO indicate that Sigma-GO is in the mGO region. In contrast, some of the observed values of NS-GO are different from those of iGO, mGO, and tGO, although the peak position of the $\pi - \pi^*$ transition of NS-GO is close to that of mGO and tGO. We do not further discuss properties of commercial GOs because of the unclear preparation method. Table 2 should be indispensable to classify GO states.

## Methods

All chemicals, except for graphite, were purchased from FUJIFILM Wako Pure Chem. Co. GO colloids were prepared from Madagascar graphite using a modified Hummers' method[50,51]. Graphite was oxidized by stirring (250 rpm) graphite (2 g) with sulfuric acid (95%, 80 mL), phosphoric acid (85%, 9 mL), and potassium permanganate (99.3%, 10 g) at $310 \pm 2$ K for 4 h. After the oxidation, 200 mL of distilled water was added slowly to the mixture, followed by 100 mL of a 1% $H_2O_2$ solution. To remove the manganese residue, the GO colloids were washed five times with hydrochloric acid (5%) and centrifuged ($10,620 \times g$). The GO colloids were finally washed five times with distilled water to obtain the clear yellow-brown GO colloids. The GO colloids were collected from the supernatant and diluted with distilled water, then divided into equal groups and ripened at different temperatures of 298, 308, 333 and 348 K for 1 to 336 h (2 weeks). We introduced the reduced time ($t_{red}$) to describe ripening processes at different temperatures based on a definite energy level of the $\pi - \pi^*$ transition band as the standard. The reduced times at different temperature were given by $t_{red}(298\,K) = 1/112 t_{298}, t_{red}(308\,K) = 1/56 t_{308}$, $t_{red}(333\,K) = 1/4 t_{333}$ and $t_{red}(348\,K) = t_{348}$. Here, $t_{TK^{-1}}$ represents the ripening time at each temperature (see Supplementary Note 1). The illuminance on the experimental desk, where the GO was prepared, was between 850 and 950 lx and the ripening was conducted under the light-shielded conditions of <5 lx by covering the flasks containing the GO colloids with Al foil. Aliquots were obtained after ripening at different durations to measure their UV-Vis absorption spectra. The ripened residual GO colloids were freeze-dried (223 K, 10 Pa) using a freeze dryer (FDU-12AS, AS-ONE). The rGO sample was obtained by reducing the non-ripened freeze-dried GO powder at 623 K for 30 min under Ar flow (100 cm³ min⁻¹). The commercial GO colloids were purchased from Sigma-Aldrich Co., LLC and Nippon Shokubai Co., Ltd. and diluted with distilled water to 0.1 wt%.

### Characterization of GO colloids and freeze-dried GO powders

The concentration of the GO colloids (0.1 wt% GO) was determined by weight loss measurements after heating at 393 K for several hours. The UV-Vis absorption spectra of the GO colloids ripened under different conditions, and commercial GO colloids were measured after 30 times dilution using an optical absorption spectrometer (JASCO Corporation V670) with distilled water as the reference. XRD patterns, Raman spectra, XPS, and magnetic moments of freeze-dried GO ripened at 348 K, commercial GOs and rGO were measured using an XRD (Rigaku Corp., SmartLab) equipped with an HyPix-3000 detector, an XPS (JEOL JPS-9010TR) and a SQUID magnetometer (Quantum Design MPMS3). The electrical resistivities of freeze-dried GO ripened at 348 K, commercial GOs and rGO were measured using a lab-made system at different compression pressures up to 240 MPa. The GO powder sample was compressed at 2 kN to produce a disc of 4 mm in diameter in advance. The contact resistance-free value was estimated from the extrapolation of the resistivity vs. the reciprocal of the compression pressure[52,53].

GO powders with highly disordered in-layer structures did not provide explicit TEM images for the detection of structural changes upon ripening. TEM images of the GO annealed at 2073 K in Ar were

**Table 1 | Average number of stacking layers determined via TEM and XRD**

| Method | GO | | |
|---|---|---|---|
| | iGO (0 h) | mGO (2 days) | tGO (7 days) |
| TEM | 14 | 9 | 6 |
| XRD | 14 | 8 | 6 |

**Table 2 | List of the important properties of GOs**

| GO | iGO | mGO | tGO | rGO[b] | Sigma | NS |
|---|---|---|---|---|---|---|
| $\pi - \pi^*$ (nm) | $230.5 \pm 0.5$ | <230.0 | > 228.8[a] | N/A | $229.6 \pm 0.2$ | $229.6 \pm 0.2$ |
| Reliability of peak deconvolution | High | Medium | Low | N/A | Medium | High |
| Oxygen (%) | $31 \pm 1$ | <30 | | 18 | 27 | 27 |
| Epoxy (%) | $33 \pm 4$ | 23–29 | <23 | 12 | 24 | 29 |
| Hydroxy (%) | $7 \pm 4$ | 13–30 | $30 \pm 2$ | 24 | 27 | 6 |
| $\chi$ ($10^{-8}$ emu g⁻¹) | 0.5–1.8 | 0–2.3 | 0.4–0.7 | 0.4 | 0.83 | 21.9 |
| Average number of stacking layers | 13–14 | 8–13 | 6–8 | N/A[c] | 12 | 8 |
| Electrical resistivity ($\Omega$ m) | >0.4 | 0.1–0.4 | <0.1 | $2 \times 10^{-6}$ | 0.31 | 2.6 |

[a]The peak position of the $\pi - \pi^*$ transition of the tGO region overlaps with those of iGO and mGO.
[b]Data for rGO obtained in this study using rGO, which is reduced iGO at 623 K under an Ar flow.
[c]The XRD pattern of rGO is completely different from those of other GO samples, as shown in Supplementary Fig. 6. The 001 peak of the rGO sample is quite small and the 002 peak is too broad to determine the accurate average number of stacking layers.
Data acquisition methods: $\pi - \pi^*$ transition, UV-Vis measurement; percentage of oxygen, epoxy group, and hydroxy group, XPS; magnetic susceptibility ($\chi$), SQUID; average number of stacking layers, XRD analysis and TEM; electrical resistivity, direct-current electrical resistivity-measurement. The reliability of peak deconvolution depends on the broadening of the $n - \pi^*$ transition peak. The reliability was evaluated from the FWHM of the $n - \pi^*$ transition peak, as follows. High reliability, FWHM < 68 nm; medium reliability, 68–78 nm; low reliability, >78 nm.

captured using a JEOL JEM-ARM-200CF with an acceleration voltage of 120 kV. EELS was conducted on a specific area of the TEM image.

## Data availability

The data that support the findings of this study are available from the corresponding author upon request.

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

## Acknowledgements

This research was funded by Toyota Motor Corporation and was partially supported by the project (JPNP14004) commissioned by the New Energy and Industrial Technology Development Organization (NEDO). Professor T. Iiyama enabled us to conduct the XPS analyses.

## Author contributions

H.O. contributed to the conceptualization, experiments, manuscript preparation, editing and review. K.U. carried out the TEM observation and EELS analyses. N.H. and T.K. carried out experiments. Y.A. carried out the magnetic property measurement and analyses. R.K. contributed to the conceptualization. T.B. supported writing the manuscript. J.U. contributed to the validation. I.M. contributed to the methodology. K.K. supervised the research.

## Competing interests

The authors declare no competing interests.
