## [Peer Review File · Nature Communications]

Conversion of graphene oxide from intrinsic state into transient stateREVIEWER COMMENTS

Reviewer #1 (Remarks to the Author):

In this work, authors provide systematic analysis of the changes in the UV-Vis absorption spectra of GO colloid solutions upon ageing (ripening) that provides new insights into the chemistry of GO colloids. By analyzing structural changes, authors single out three stages of GO ripening in colloid solutions, leading to consecutive formation of three states of GO, named as intrinsic, metastable and transient. Based on the XPS analysis, authors further analyze how the content of different functional groups change with time. The TEM, XPS, and XRD methods are also employed.

This is a good work, adding to understanding the GO chemistry.

I found only two omissions.

1) It is unclear how the crystallite size was determined from the XRD data. The original XRD pattern and respective calculations should be shown. Also, it is probably not fully correct to use the term "crystallite size" toward what authors determine along the c-axis.

2) I am doubtful that the crystallite size shown in Fig. S8 can be determined from the Raman data. First off all, the original Raman spectra are missing. Secondly, and most importantly, the lateral density of the point defects in original GO is significantly above the level to be detected based on the ID/IG ratio. And this density of the defects does not decrease with ripening. I would remove the Raman part from the manuscript.

However, I believe these omissions do not affect the main conclusions made in this work, and can be easily fixed by the authors in the course of minor revision.

Some technical details that can be improved.

1) In my version of the manuscript the $n-n$ and $n-n$ formulas in the text are not properly shown. This strongly complicated the reading.

2) Fig. 1b,c. The lines overlap, the actual change in the spectra in time is hardly visible. Make an inset, showing the enlarged fragment in the 200–280 nm range for better resolution. In Fig 1c, an enlargement of the 300 nm region is also desired.

Reviewer #2 (Remarks to the Author):

This is a well written paper on an important topic. The structure of GO is complex and in solution it is typically unstable, making experiments difficult to reproduce. Here the authors study the thermal and temporal evolution of the material using a variety of optical, TEM, and, for freeze-dried samples, magnetic probes. The experiments appear to have been carefully carried out and the conclusions drawn are scientifically sound.

The authors also provide methods for stabilization of the material, for example by storing GO dispersions below 255K or by adding ammonium peroxydisulfate. These prescriptions are bound to be highly useful to avoid unwanted changes and to preserve "standardized" materials for scientific study or application.

One shortcoming in the paper is inadequate discussion of what constitutes the "ripening" process. As correctly stated in the introduction, the structure of GO can be sensitive to time beyond initial preparation, coupled to storage temperature, AS WELL AS photon-induced changes. The experiments presented here keep good track of time and temperature, but (unless I missed it), nowhere is it clearly stated what the light exposure was. Were the samples subjected to "room light" throughout? If so, what is room light? What about sunlight? Or were the beakers containing the samples always covered with metal foil to prevent any possible contamination from ambient light? Is storage at 255K done in the dark or with light exposure? Does it make a difference?

I think these concerns need to be addressed directly in the paper. However, I'm inclined to say that, with suitable revision/clarification, the paper is appropriate for publication in Nature Communications.

Reviewer #3 (Remarks to the Author):

The authors explore the changes induced in graphene oxide over time and by raising the temperature. Based on UV-Vis spectroscopy results, they propose three regimes for the evolution of graphene oxide: intrinsic, metastable and transient states. The materials are studied by XRD, Raman spectroscopy, XPS, TEM, EELS and they provide magnetic measurements.

Indeed, a study of the characteristics of graphene oxide is necessary to standardize GO science and applications. However, several points need to be addressed before rendering the study suitable for publication, I recommend acceptance with major changes.

1. In the abstract, the authors do not mention the complete characterization included in the study, only optical measurements are cited. It is worth mentioning the characterization techniques performed that support the conclusions of the authors (XRD, Raman, XPS, TEM, EELS, magnetic properties) and to provide a clear picture of the study to the reader.

2. The UV-Vis characterization is the first mentioned, however many conclusions are drawn in the first lines when the results are exposed without proper evidence, for example:

Lines 111, 112 The authors state "Ripening at 348 K resulted in a significant change in the n conjugated structure, which is associated with a considerable change of OFGs." At this point no evidence proves that the OFGs change at this temperature

Line 146 The authors state "the three regions correspond to three structurally different GO states, which are unlike rGO." Nevertheless, the structure of the GO states has not been discussed, nor the structure of rGO

Line 154 "..... is ascribed to transient GO (tGO) as the GO colloids transforming into rGO" The transformation or evolution of tGO into rGO has not been properly evidenced at this point. Do the authors claim that all tGO turns into rGO? Where is the line drawn from tGO to rGO? what are the main differences between the tGO state and the rGO state?

Line 169 "...which come from the partial detachment of the surface functional groups and growth of the graphene-like structure" this conclusion may be supported later on by the XRD and XPS characterization, however at this point this conclusion is not solidly supported.

3. Add details on the calculation of the reduced time course (t_r) in the supporting information. This parameter leads to graph 3, which sets the bases for the classification proposed in this study.

4. Place figures in the supporting information file in the order they are mentioned in the manuscript, since the Raman figure is S8 and is mentioned right after XRD results.

5. Add XRD and Raman spectra in the supporting information file

6. Sp²/sp³ ratio can also be calculated from XPS spectra, cross check EELS results with XPS to validate results

7. Electric properties are also indicative of the "type" of graphene oxide, include electric properties (resistivity, conductivity) of the different GO states (iGO, mGO, tGO and rGO)

8. The classification proposed by the authors (based on figure 3) is valuable, however beyond the labeling, a full set of parameters based on the characterization performed should be specified to be able to classify the GO material. For example, in order to be labeled as iGO, what color should the aqueous sample be? What should be its absorption? What should be its interlayer distance and its Lc? What should be the sp²/sp³ ratio? What functional groups should be present and in what proportion? A set of value ranges for each parameter would help to classify the GO sample into the categories proposed by the authors.

Is the goal of the study that researchers/engineers would be able to classify GO merely on the results of UV-VIS? if so this should be properly highlighted and justified.

Title: Understanding the ever-changing nature of graphene oxide: Intrinsic, metastable and transient states

Hayato Otsuka, Koki Urita, Nobutaka Honma, Takashi Kimuro, Yasushi Amako,
Radovan Kukobat, Teresa Bandosz, Junzo Ukai, Isamu Moriguchi, Katsumi Kaneko

Manuscript Ref. No. NCOMMS-23-20205

Response to Reviewer's comments:

At first, we appreciate your consideration of our manuscript (NCOMMS-23-20205). We have considered Reviewers' suggestions carefully to revise in the manuscript (Highlighted in yellow color).

Detailed point-by-point responses to the reviewers' comments are attached below. Here, the comments and questions from the reviewers are written in red and our answers are given in blue.

Reviewer #1

Thank you very much for the helpful comments that, which have significantly improved the quality of our manuscript. We revised the manuscript considering all your comments.

Comment 1: It is unclear how the crystallite size was determined from the XRD data. The original XRD pattern and respective calculations should be shown. Also, it is probably not fully correct to use the term "crystallite size" toward what authors determine along the c-axis.

Answer 1: We apologize for the confusion. The original XRD patterns (Supplementary Fig. 7) and the determination methods are presented in the Supplementary Information. We have also revised the expression of the crystallite size to the average number of stacking layers for clarity. As the terms "stack height" and "stack width" are commonly used for the disordered carbon materials, we used these expressions. We have revised the manuscript as follows:

lines 239-247

"Fig. 6a shows the changes in the average number of layers and the interlayer distance of GOs and rGO from the X-ray diffraction (XRD) (ref. Supplementary Fig. 7). The average number of stacking

layers was calculated from the stack height (L_c) and the interlayer distance. Details are presented in the Supplementary Information (see Supplementary Note 2). The average number of stacking layers decreases monotonically upon ripening, suggesting spontaneous exfoliation of GO in an aqueous media. The initial number of stacking layers of the non-ripened GO is 14, which becomes 6 after ripening for 168 h. All three GO states undergoes exfoliation. The interlayer distance is unchanged in the iGO and mGO regions but decreases to 0.74 nm in the tGO region after ripening for 168 h, which is close to that of rGO (0.71 nm).”

We have added the XRD patterns and an explanation of the determination method and the terminology in the Supplementary Information.

Supplementary Fig. 7 | XRD patterns of the freeze-dried GO ripened for different durations at 348 K and rGO.

Supplementary Note 2 | Determination and terminology of in-plane size and thickness along stacking direction.

Graphene oxide has the unit structure consisting of several oxidized graphene-like sheets. The size of the oxidized graphene-like sheet and the thickness of the stacking sheets along the c-axis are called the “stack width” and “stack height”, respectively. These expressions are often used for micrographite unit structures of disordered carbon. The stack width and stack height are expressed as L_a and L_c , respectively. In this paper, we use the average number of stacking layers instead of the stack height for showing the stacking structure more clearly. The average number of stacking layers is calculated as follows

$$\text{Average number of stacking layers} = \frac{L_c}{d_{001}} + 1$$

L_c is determined from the Scherrer equation using the full width half maximum (FWHM) of the 001 peak of GO and d_{001} , i.e. the interlayer distance of the (001) planes based on the Bragg equation. The broadening of the peak caused by instruments is corrected by subtracting the FWHM of the graphite 002 peak, which has a sufficient crystallite size.

$$\text{FWHM}_{\text{GO-001}} = \sqrt{(\text{FWHM}_{\text{GO-001-observed}}^2 - \text{FWHM}_{\text{Graphite-002}}^2)}$$

$$L_c = 0.94 \times \frac{0.15418}{\frac{\text{FWHM}_{\text{GO-001}}}{180} \pi \times \cos\left(\frac{2\theta}{360} \pi\right)}$$

Here, the 001 peaks of most of the samples are not of λ -shape. Then, we use the shape factor of 0.94, which is the shape factor for the three-dimensional structure. The wavelength of the X-rays is 0.15418 nm and θ denotes the position of the 001 peak.

Comment 2: I am doubt that the crystallite size shown in Fig. S8 can be determined from the Raman data. First of all, the original Raman spectra are missing. Secondly, and most importantly, the lateral density of the point defects in original GO is significantly above the level to be detected based on the I_D/I_G ratio. And this density of the defects does not decrease with ripening. I would remove the Raman part from the manuscript. However, I believe these omissions do not affect the main conclusions made in this work, and can be easily fixed by the authors in the course of minor revision.

Answer 2: Thank you for your valuable comments. As you mentioned, the lateral density of the point defects, i.e. the “stack width”, is significantly above the level to be detected using the I_D/I_G ratio. We have recalculated the I_D/I_G ratio not from the peak intensities but from the peak areas of the D band and G band after peak deconvolution. An example of peak deconvolution is shown in Supplementary Fig. 8b. The newly obtained results were largely unchanged from those of the previous analysis; however, a slight change in the stack width with ripening was observed. The stack width slightly increased in the iGO and mGO states but slightly decreased in the tGO state, suggesting the structural change of the in-plane structure with ripening. The change in the stack width is attributed to the change in the OFGs. However, this change is not significant, as you suggested. We need to study Raman spectroscopic data from another approach in a future work.

As you recommended, the omission of the Raman part is one possibility to simplify the discussion of the ripening process. However, another reviewer requested us to show the original Raman spectra. Thus, we did not remove the Raman spectroscopic data giving the slight description.

In response to your comment, we have revised our manuscript and added an explanation for the calculation of the stack width from the I_D/I_G ratio in the Supplementary Information as follows.

lines 247-253

“The stack width of the ripened GOs and rGO was determined by Raman spectroscopy [42,43]. The Raman spectra are shown in Supplementary Fig. 8a and the stack width was determined from the I_D/I_G ratio derived using the peak areas after peak deconvolution. An example of the peak deconvolution is shown in Supplementary Fig. 8b and the derivation of the stack width is presented in detail in the Supplementary Information (see Supplementary Note 2). The stack width does not change throughout

the ripening period, except for a slight increase near the boundary between the mGO and tGO states (Supplementary Fig. 8c).”

Supplementary Information, p10 in Supplementary Note 2

“The L_a from the Raman spectrum is determined according to the I_D/I_G ratio using the Tuinstra-Koenig^{1,2)} relation.

$$\frac{I_D}{I_G} = \frac{(2.4 \times 10^{-10})\lambda_l^4}{L_a}$$

The I_D/I_G ratio is determined from the peak area after peak deconvolution. An example of the peak deconvolution is shown in Supplementary Fig. 8b. Although this relation is typically used for graphite and carbon nanotube, it has been often applied to amorphous carbon¹⁾ and graphene oxide²⁾.

¹⁾ Ferrari, A. C. & Robertson, J. Interpretation of Raman spectra of disordered and amorphous carbon. *Phys. Rev. B*, **61**, 14095-14107 (2000).

²⁾ Mattevi, C. Eda, G. Agnoli, S. Miller, S. Mkhoyan, K. A. Celik, O. Mastrogiovanni, D. Granozzi, G. Garfunkel, E. & Chhowalla M. Evolution of electrical, chemical, and structural properties of transparent and conducting chemically derived graphene thin films. *Adv. Funct. Mater.*, **19**, 2577-2583 (2009).”

Additionally, we have added the original Raman spectra and an example of peak deconvolution in Supplementary Figs. 8a and b and redrawn the plot of the stack width as a function of the ripening time in Supplementary Fig. 8c.

Supplementary Fig. 8 | Changes in the Raman spectra and the stack width. **a** Raman spectra of the freeze-dried GO ripened for different durations at 348 K and the rGO sample. **b** Example of peak deconvolution of the Raman spectrum of the freeze-dried GO ripened at 348 K for 48 h. **c** Change in the stack width determined from a Raman spectrum. The stack width of the GOs ripened at 348 K is represented with black circles. The stack width of rGO is shown for comparison (brown square).

Technical comment 1: In my version of the manuscript the π - π and n- π formulas in the text are not properly shown. This strongly complicated the reading.

Technical comment 2: Fig. 1b,c. The lines overlap, the actual change in the spectra in time is hardly visible. Make an inset, showing the enlarged fragment in the 200-280 nm range for better resolution. In Fig 1c, an enlargement of the 300 nm region is also desired.

Answers for the technical comments

Thank you for your comments. We revised our manuscript as follows.

- 1) Considering your comments, we converted the π - π^* and n- π^* formulas from the “Symbol” forms to the equation form to avoid the garbled characters throughout the manuscript and the Supplementary Information.
- 2) We added the magnified peaks in the corresponding wavelength regions in the insets (Figs. 1b and c). We also improved **Supplementary Fig. 1 e-l** for clarity.

Fig. 1 | Changes in the optical properties of GO colloids. **a** Photographs of the GO colloids ripened at different temperatures and for different durations. **b, c** UV-Vis absorption spectra of GO colloids ripened at 298 K (**b**) and 348 K (**c**) for different durations. The ripening time: black, 0 h; brown, 1 h; purple, 3 h; blue, 7 h; pale blue, 12 h; yellow-green, 24 h; green, 48 h; orange, 120 h (5 days) and red, 168 h (7 days).

Supplementary Fig. 1 | Changes in the UV-Vis spectra for the GO colloids as a function of the ripening time at different temperatures. a-d Full range of the UV-Vis spectra. e-h Magnified spectra in the range of 200-260 nm. i-l Magnified spectra in the range of 280-300 nm.

Reviewer #2

Comment 1: One shortcoming in the paper is inadequate discussion of what constitutes the "ripening" process. As correctly stated in the introduction, the structure of GO can be sensitive to time beyond initial preparation, coupled to storage temperature, AS WELL AS photon-induced changes. The experiments presented here keep good track of time and temperature, but (unless I missed it), nowhere is it clearly stated what the light exposure was. Were the samples subjected to "room light" throughout? If so, what is room light? What about sunlight? Or were the beakers containing the samples always covered with metal foil to prevent any possible contamination from ambient light? Is storage at 255K done in the dark or with light exposure? Does it make a difference?

Answer 1: Thank you for your important comments. We neglected to describe the experimental conditions of ripening. We conducted the ripening process under the light-shielded conditions by covering the flasks containing the GO colloids with Al foil. However, we must describe quantitatively the photo illumination conditions, as you pointed out. We measured the illuminance at the experimental desk and the inside the reaction flask covered with Al foil.

We added quantitative explanations on the light irradiation in the Method section in lines 339-342.

“The illuminance on the experimental desk, where the GO was prepared, was between 850 and 950 lx and the ripening was conducted under the light-shielded conditions of <5 lx by covering the flasks containing the GO colloids with Al foil.”

Additionally, storage at 255 ± 2 K was conducted in the more light-shielded condition using a freezer. We added an explanation (lines 290-292), as follows.

“The most effective preservation method is storage at 255 ± 2 K in the frozen state in the more light-shielded conditions ($\ll 5$ lx) (Supplementary Fig. 11), which guarantees the stability of iGO for at least 1 month.”

Reviewer #3

Comment 1: In the abstract, the authors do not mention the complete characterization included in the study, only optical measurements are cited. It is worth mentioning the characterization techniques performed that support the conclusions of the authors (XRD, Raman, XPS, TEM, EELS, magnetic properties) and to provide a clear picture of the study to the reader.

Answer 1: Thank you for your valuable comment. We have added other characterization methods to support our conclusions.

lines 34-37

“The presence of three states of GO is supported by the compositional changes of oxygen functional groups detected via X-ray photoelectron spectroscopy, magnetic property measurements and structural information from X-ray diffraction and transmission electron microscopy observation.”

Comment 2: The UV-Vis characterization is the first mentioned, however many conclusions are drawn in the first lines when the results are exposed without proper evidence, for example:

Lines 111, 112 The authors state “Ripening at 348 K resulted in a significant change in the π conjugated structure, which is associated with a considerable change of OFGs.” At this point no evidence proves that the OFGs change at this temperature

Line 146 The authors state “the three regions correspond to three structurally different GO states, which are unlike rGO.” Nevertheless, the structure of the GO states has not been discussed, nor the structure of rGO

Line 154 “..... is ascribed to transient GO (tGO) as the GO colloids transforming into rGO” The transformation or evolution of tGO into rGO has not been properly evidenced at this point. Do the authors claim that all tGO turns into rGO? Where is the line drawn from tGO to rGO? what are the main differences between the tGO state and the rGO state?

Line 169 “....which come from the partial detachment of the surface functional groups and growth of the graphene-like structure” this conclusion may be supported later on by the XRD and XPS characterization, however at this point this conclusion is not solidly supported.

Answer 2: Thank you for your helpful comments. We have revised the manuscript and added several sentences to support our conclusions.

lines 117-120

“Ripening at 348 K results in a significant change in the π -conjugated structure, which is associated with the compositional change of OFGs as discussed below. The changes upon ripening at 348 K are discussed with a focus on OFG changes induced by modification of the π -conjugated structure there.”

lines 154-156

“The GO corresponding to three states should have individual π -conjugated electronic structures, as the $\pi - \pi^*$ transition is assigned to the electronic transition of the conjugated frame structure of GO.”

lines 162-164

“The red-shift region, which is only observed in the GO colloids ripened at 348 K for longer than 40 h, is attributed to transient GO (tGO) as the GO colloids transforming into the reduced form of GO.”

lines 178-180

“... absorption band intensities of the $\pi - \pi^*$ and $n - \pi^*$ transitions, which is related to the partial detachment of the surface functional groups and growth of the graphene-like structure, as mentioned in the Discussion.”

lines 308-311

“The reduction of GO starts in the tGO region, as indicated by a reduction in the oxygen content, which eliminates defects and thereby decreases the magnetic susceptibility. The three regions correspond to three structurally different GO states, which are completely different from rGO.”

The difference between tGO and rGO can be observed in the electrical properties. The electrical properties of GO and rGO have been added in Supplementary Fig. 6. The electrical resistivity of rGO is 4 orders of magnitude lower than that of tGO, as mentioned in Answer 7. The XPS analysis also provides information on the differences between tGO and rGO. The oxygen content (Fig. 4c) of rGO is lower than that of tGO.

Fig. 4 | Changes in the structure and OFGs with the reduced time course. The horizontal axes are expressed as the logarithm of the ripening time of the GO colloids ripened at 348 K. **a, b** Band intensity changes of iGO, mGO and tGO with the ripening time. **a** $\pi - \pi^*$ transitions. **b** $n - \pi^*$ transitions. Ripening temperature: \circ , 298 K; \square , 308 K; \diamond , 333 K and \blacktriangle , 348 K. Band intensities of the non-ripened GO colloid are shown with \bullet . **c** Total oxygen content determined by XPS. **d** Compositional changes of the oxygen-containing functional groups determined from the deconvoluted C1s spectra. The fraction of carbon atoms not bonded to oxygen atoms is excluded.

Comment 3: Add details on the calculation of the reduced time course (t_r) in the supporting information. This parameter leads to graph 3, which sets the bases for the classification proposed in this study.

Answer 3: Thank you for this valuable comment. We added details regarding the derivation of the reduced time (t_r) in the Supplementary Information.

Supplementary Note 1 | Definition and significance of reduced time

We introduce the reduced time to describe ripening processes at different temperatures based on a definite energy level as the standard. The peak position of the $\pi - \pi^*$ transition band depends on the ripening time and temperature. If the ripening processes at different temperatures proceed in the same way we can describe the time changes of the peak position at different temperatures with a single relation by introducing the optimum reduced time (t_r). The peak position varies from 230.4 to 228.8 nm for UV-Vis spectra of iGO and mGO. We select the intermediate energy level (229.6 nm) between these positions as the standard energy level. Then, we determine the ripening time for shifting of the peak position to 229.6 nm at different temperatures as follows. The times for shifting to 229.6 nm are 336 h at 298 K, 168 h at 308 K, 12h at 333 K, and 3 h at 348 K. The reduced time at different temperature is obtained, as given below.

$$t_r(298 \text{ K}) = t_{298} \times \frac{3 \text{ h}}{336 \text{ h}} = \frac{1}{112} t_{298}$$

$$t_r(308 \text{ K}) = t_{308} \times \frac{3 \text{ h}}{168 \text{ h}} = \frac{1}{56} t_{308}$$

$$t_r(333 \text{ K}) = t_{333} \times \frac{3 \text{ h}}{12 \text{ h}} = \frac{1}{4} t_{333}$$

$$t_r(348 \text{ K}) = t_{348} \times \frac{3 \text{ h}}{3 \text{ h}} = t_{348}$$

Here, $t_{T/K}/h$ represents the ripening time at each temperature.

Comment 4: Place figures in the supporting information file in the order they are mentioned in the manuscript, since the Raman figure is S8 and is mentioned right after XRD results.

Answer 4: Thank you for pointing out our mistakes. We have changed the order of the figures in the Supplementary Information to match the description in the manuscript.

Comment 5: Add XRD and Raman spectra in the supporting information file

Answer 5: We have added the XRD patterns and Raman spectra in **Supplementary Figs. 7 and 8a**, respectively.

Supplementary Fig. 7 | XRD patterns of the freeze-dried GO ripened for different durations at 348 K and rGO.

Supplementary Fig. 8 | Changes in the Raman spectra and the stack width. **a** Raman spectra of the freeze-dried GO ripened for different durations at 348 K and the rGO sample. **b** Example of peak deconvolution of the Raman spectrum of the freeze-dried GO ripened at 348 K for 48 h. **c** Change in the stack width determined from a Raman spectrum. The stack width of the GOs ripened at 348 K is represented with black circles. The stack width of rGO is shown for comparison (brown square).

Comment 6: Sp²/sp³ ratio can also be calculated from XPS spectra, cross check EELS results with XPS to validate results

Answer 6: Thank you for pointing this out. We created a plot of the sp²/sp³ ratios obtained from XPS and EELS, as shown below.

Figure R1 | Correlation diagram of the sp²/sp³ ratios determined via XPS and EELS.

The sp²/sp³ ratio determined from the EELS spectrum is far lower than that determined from the XPS spectrum. This is attributed to the difference in the observed position. The EELS spectrum was measured on the edge part of GO, while the XPS spectrum provides bulk information. The edge part of GO is considered to be highly oxidized and have a low sp²/sp³ ratio even after the thermal reduction at 2073 K. Moreover, the correlation diagram indicates a negative correlation, suggesting that the ripening processes occurring in the edge and bulk positions differ. This will be clarified in future studies. We added the explanation for the difference between the sp²/sp³ ratios determined from the EELS and XPS spectra in the manuscript.

lines 273-274

“Here, the sp²/sp³ ratio obtained from the EELS analysis is significantly lower than that obtained from the XPS analysis because the EELS spectra were measured only at the edge part of GO.”

Comment 7: Electric properties are also indicative of the “type” of graphene oxide, include electric properties (resistivity, conductivity) of the different GO states (iGO, mGO, tGO and rGO)

Answer 7: Thank you for your suggestion. We measured the electrical resistivities of the GO samples prepared from the freeze-dried GO powders of iGO, mGO, tGO and commercial GOs. As you suggested, the electrical resistivity is an efficient way to distinguish the different GO states. We have added the resistivity data in **Supplementary Fig. 6** and added an explanation in the manuscript.

lines 224-236

“The electrical properties of GO are closely related to the π -conjugated electronic structure and the composition of OFGs, which should support the presence of three GO states. The direct-current electrical resistivities of pelletized GO samples were measured using the two-electrode method under various compression pressures. The pelletized GO specimen was prepared from freeze-dried powder of the iGO, mGO, tGO and commercial GOs. Supplementary Fig. 6 presents the electrical resistivities of GO samples in different states as a function of the reciprocal of the compression pressure. The data of rGO and commercial GOs are also shown, for comparison. The electrical resistivity of iGO increases with the compression pressure, which is different from mGO and tGO. The compression of mGO and tGO improves the interparticle contact and then their electrical resistivities slightly increase with the compression pressure. The absolute electrical resistivities of iGO, mGO and tGO are much higher than that of the electrically conductive rGO. The reverse tendency of the electrical resistivity of iGO should be associated with the flexible nature^[41], although we must study this point more quantitatively in future.”

Supplementary Fig. 6 | Electrical resistivities of pelletized GO samples measured via the two-electrode method. a Resistivity as a function of the reciprocal of the compression pressure. **b** Electrical resistivity change of the freeze-dried GO ripened at 348 K and rGO. The resistivities were measured under applied pressure in the range of 10–240 MPa. Here, we extrapolated the plot of the resistivity vs. reciprocal compression pressure P_c to the ordinate ($P_c = \infty$) to obtain the contact-resistance minimum value, even though the resistivity does not markedly depend on P_c . The resistivity at $P_c = \infty$ gradually decreases with ripening and drops for rGO, indicating a significant difference between iGO and rGO. Sample specifications: iGO, non-ripened GO; mGO, GO ripened at 348 K for 12 h; tGO, ripened at 348 K for 168 h; rGO, freeze-dried GO reduced at 623 K for 30 min. The resistivity values at $P_c = \infty$ for different GO samples and commercial ones are presented in Table S2.

Comment 8: The classification proposed by the authors (based on figure 3) is valuable, however beyond the labeling, a full set of parameters based on the characterization performed should be specified to be able to classify the GO material. For example, in order to be labeled as iGO, what color should the aqueous sample be? What should be its absorption? What should be its interlayer distance and its Lc? What should be the sp²/sp³ ratio? What functional groups should be present and in what proportion? A set of value ranges for each parameter would help to classify the GO sample into the categories proposed by the authors.

Is the goal of the study that researchers/engineers would be able to classify GO merely on the results of UV-VIS? if so this should be properly highlighted and justified.

Answer 8: Thank you for your valuable comment. Surely, the observed values from different characterization methods from UV-Vis spectroscopy, XPS, SQUID, XRD, TEM and electrical resistivity measurement are effective to classify the GO state. We summarize the values in Supplementary Table. 2. Here, we adopt the peak position of the $\pi - \pi^*$ transition to exhibit precisely the color difference. The peak position of the $\pi - \pi^*$ transition band of the GO colloids is the most effective and easy way to distinguish iGO from mGO and tGO. Additionally, other properties are helpful for identifying the GO state.

Table S2 A list of the important properties of GOs.

GO Properties	iGO	mGO	tGO	Sigma	NS
π - π^* (nm) (UV-Vis)	230.5 \pm 0.5	< 230.0	> 228.8*	229.6 \pm 0.2	229.6 \pm 0.2
Reliability of peak deconvolution	High	Medium	Low	Medium	Meduim
Oxygen% (XPS)	31 \pm 1	< 30		27	27
Epoxy% (XPS)	33 \pm 4	23 ~ 29	< 23	24	29
Hydroxyl% (XPS)	7 \pm 4	13 ~ 30	30 \pm 2	27	6
χ (10^{-8} emu g ⁻¹) (SQUID)	0.5 ~ 1.8	0 ~ 2.3	0.4 ~ 0.7	0.83	21.9
Average number of stacking layers (XRD & TEM)	13 ~ 14	8 ~ 13	6 ~ 8	12	8
L_a (nm) (Raman spectroscopy)	12 ~ 14			12 \pm 1	12 \pm 1
Electrical resistivity (Ω m)	> 0.4	0.1 ~ 0.4	< 0.1	0.31	2.6

The peak position of the $\pi - \pi^$ transition of iGO and mGO is overlapped with that of the tGO region.

Supplementary Note 3 | Reliable classification.

The color of tGO differs significantly from those of iGO and mGO, as shown in Fig. 3 and Supplementary Fig. 3, although the peak position of the $\pi - \pi^*$ transition is the same. Moreover, the UV-Vis spectra of tGO are completely different from those of iGO and mGO. As the peaks are very broad and the peaks of the $\pi - \pi^*$ and $n - \pi^*$ transitions overlap significantly, it is difficult to reliably determine the peak position. Thus, the UV-Vis spectra measurement is the most effective for classification of three GOs. In conclusion, comparative characterization with XPS, XRD analysis, TEM and measurements of magnetic and electrical properties in addition to the UV-Vis spectroscopic examination can provide the most reliable classification of iGO, mGO and tGO.

The color is difficult to express quantitatively; thus, we have selected the optical absorption to identify the GO colloids owing to their slight peak position difference. Thus, we must mention the necessity of the UV-Vis spectrum of the GO colloid. Although XRD and XPS analyses are powerful and informative techniques for classifying the GO state,

these measurements must be performed after drying of the GO colloids and required calibrations. In comparison, the determination of the peak positions of the UV-Vis spectrum of the GO colloids is simple without changing colloidal state of GO. We added the classification method for distinguishing iGO from mGO and tGO to the Discussion section.

lines 317-328

“We summarize key properties of iGO, mGO and tGO, which are applicable to their classification in Supplementary Table 2. Those properties of commercial GOs are shown in the table for comparison. Supplementary Table 2 shows that various properties are helpful for distinguishing iGO from mGO and tGO (see Supplementary Note 3). In particular, the peak position of the $\pi - \pi^*$ transition band of the GO colloids is the most effective and easy way to identify iGO. The information on OFGs from XPS, the magnetic susceptibility, the number of stacking layers based on XRD and TEM and the electrical resistivities of three GO states are mutually different each other. Here, most of the observed values from various methods of Sigma-GO indicate that Sigma-GO is in the mGO region. In contrast, some of the observed values of NS-GO are different from those of iGO, mGO and tGO, although the peak position of the $\pi - \pi^*$ transition of NS-GO is close to that of mGO and tGO. We do not further discuss properties of commercial GOs because of the unclear preparation method. Supplementary Table 2 should be indispensable to classify GO states.”

REVIEWER COMMENTS

Reviewer #1 (Remarks to the Author):

Authors have addressed one out of two main questions, raised during the first submission. However, the second question remained not resolved. Moreover, by trying to address my question and to simultaneously please reviewer 3, the authors made the situation even worse than it was before. This is why, this time I must request major revision.

Again, this is a very good work, and my opinion about this work in general has not changed. This is why I do not want this nice work to be spoiled by an incorrect use of the Raman data. Based on the authors answer to my question, I have realized that they had not understood my main concern, expressed in my first review. Here is the track of the questions and answers. I highlight the most critical points.

My previous question: I am doubt that the crystallite size shown in Fig. S8 can be determined from the Raman data. First of all, the original Raman spectra are missing. Secondly, and most importantly, the lateral density of the point defects in original GO is significantly above the level to be detected based on the ID/IG ratio. And this density of the defects does not decrease with ripening. I would remove the Raman part from the manuscript. However, I believe these omissions do not affect the main conclusions made in this work, and can be easily fixed by the authors in the course of minor revision.

Authors Respond: Thank you for your valuable comments. As you mentioned, the lateral density of the point defects, i.e. the "stack width", is significantly above the level to be detected using the ID/IG ratio.

My comment. Authors misinterpret my words. I have never mentioned that that the lateral density of the point defects is the same as the stack width. Please, see above. This is the main mistake the authors make in this work.

... The stack width slightly increased in the iGO and mGO states but slightly decreased in the tGO state, suggesting the structural change of the in-plane structure with ripening.

...As you recommended, the omission of the Raman part is one possibility to simplify the discussion of the ripening process. However, another reviewer requested us to show the original Raman spectra. Thus, we did not remove the Raman spectroscopic data giving the slight description. In response to your comment, we have revised our manuscript and added an explanation for the calculation of the stack width from the ID/IG ratio in the Supplementary Information as follows.

My comment. The calculation of the stack width based on the I(D)/I(G) ratio, authors made, is INCORRECT. This is wrong in principle. The stack width for GO cannot be derived from the Raman spectra, and especially from the I(D)/I(G) ratio.

The La value, in the Tuinstra equation indeed represents the crystallinity of graphite. However, this value is not for the thickness of the crystallite, i.e. not for the number of layers in the stack (as authors assume), but rather for the lateral size of the crystallite. The Tuinstra equation is absolutely not applicable toward GO, because GO does not have crystallite grains as graphite does. With respect to graphene and GO the I(D)/I(G) ratio represents the distance between the two neighboring point defects within one graphene plane, not the number of layers in the stack. And the density of the point defects within one graphene plane in GO significantly exceeds the threshold level to be assessed by the I(D)/I(G) ratio. Importantly, in GO this density of defects does not change with GO reduction/modification.

Once again, let me stress the main point. Raman spectroscopy is not informative toward GO. This is because the density of the defects in GO is way above the threshold values, when it can affect the I(D)/I(G) ratio. This is well known for the experts in the field. Here are some publications on this topic by the recognized experts.

Quantifying Defects in Graphene via Raman Spectroscopy at Different Excitation Energies <https://pubs.acs.org/doi/10.1021/nl201432g>

Quantifying ion-induced defects and Raman relaxation length in graphene <https://www.sciencedirect.com/science/article/abs/pii/S0008622310000138>

Unfortunately, using Raman toward GO is currently the widespread mistake, made by researchers not well familiar with the subject. However, as an expert, I cannot support the publication, where obviously wrong things are stated.

Reviewer #2 (Remarks to the Author):

The authors have revised the manuscript to my satisfaction. I recommend publication in Nature Communications.

Reviewer #3 (Remarks to the Author):

The authors have addressed the totally of the points raised on my previous revision. I think that the addition of the electrical properties strengthened the manuscript and supported the conclusions made from other characterization techniques.

I have three additional, minor comments on this revised version of the manuscript:

1. Homogenize the symbols used in figure 2 so that the symbology remains constant for figures 2,3, 4...etc
2. Merge supplementary Figure 6 with Figure 5 so that a figure of magnetic and electrical properties is included in the main manuscript.
3. Migrate Supplementary Table 2 to the main manuscript. It is my opinion that this table summarizes important characteristics of the studied materials and that important conclusions are drawn from it, therefore its inclusion in the main text is important. Moreover, include a column of rGO in this table to compare it with iGO, mGO and tGO.

**Title: Understanding the ever-changing nature of graphene oxide:
Intrinsic, metastable and transient states**

Hayato Otsuka, Koki Urita, Nobutaka Honma, Takashi Kimuro, Yasushi Amako,
Radovan Kukobat, Teresa Bandosz, Junzo Ukai, Isamu Moriguchi, Katsumi Kaneko

Manuscript Ref. No. NCOMMS-23-20205A

Response to Reviewer's comments:

Thank you so much for valuable comments and recommendations to our paper. We have considered Reviewers' comments and suggestions carefully to revise in the manuscript (Highlighted in yellow color).

Detailed point-by-point responses to the reviewers' comments are attached below. Here, the comments and questions from the reviewers are written in red and our answers are given in blue.

Reviewer #1

Comment 1:

Authors have addressed one out of two main questions, raised during the first submission. However, the second question remained not resolved. Moreover, by trying to address my question and to simultaneously please reviewer 3, the authors made the situation even worse than it was before. This is why, this time I must request major revision.

Again, this is a very good work, and my opinion about this work in general has not changed. This is why I do not want this nice work to be spoiled by an incorrect use of the Raman data. Based on the authors answer to my question, I have realized that they had not understood my main concern, expressed in my first review. Here is the track of the questions and answers. I highlight the most critical points.

My previous question: I am doubt that the crystallite size shown in Fig. S8 can be determined from the Raman data. First of all, the original Raman spectra are missing. Secondly, and most importantly, the lateral density of the point defects in original GO is significantly above the level to be detected based on the ID/IG ratio. And this density of the defects does not decrease with ripening. I would remove the Raman part from the manuscript. However, I believe these omissions do not affect the main conclusions made in this work, and can be easily fixed by the authors in the course of minor revision.

Authors Respond: Thank you for your valuable comments. As you mentioned, the lateral

density of the point defects, i.e. the “stack width”, is significantly above the level to be detected using the ID/IG ratio.

My comment. Authors misinterpret my words. I have never mentioned that that the lateral density of the point defects is the same as the stack width. Please, see above. This is the main mistake the authors make in this work.

... The stack width slightly increased in the iGO and mGO states but slightly decreased in the tGO state, suggesting the structural change of the in-plane structure with ripening.

...As you recommended, the omission of the Raman part is one possibility to simplify the discussion of the ripening process. However, another reviewer requested us to show the original Raman spectra. Thus, we did not remove the Raman spectroscopic data giving the slight description. In response to your comment, we have revised our manuscript and added an explanation for the calculation of the stack width from the ID/IG ratio in the Supplementary Information as follows.

My comment. The calculation of the stack width based on the I(D)/I(G) ratio, authors made, is INCORRECT. This is wrong in principle. The stack width for GO cannot be derived from the Raman spectra, and especially from the I(D)/I(G) ratio.

The La value, in the Tuinstra equation indeed represents the crystallinity of graphite. However, this value is not for the thickness of the crystallite, i.e. not for the number of layers in the stack (as authors assume), but rather for the lateral size of the crystallite. The Tuinstra equation is absolutely not applicable toward GO, because GO does not have crystallite grains as graphite does. With respect to graphene and GO the I(D)/I(G) ratio represents the distance between the two neighboring point defects within one graphene plane, not the number of layers in the stack. And the density of the point defects within one graphene plane in GO significantly exceeds the threshold level to be assessed by the I(D)/I(G) ratio. Importantly, in GO this density of defects does not change with GO reduction/modification.

Once again, let me stress the main point. Raman spectroscopy is not informative toward GO. This is because the density of the defects in GO is way above the threshold values, when it can affect the I(D)/I(G) ratio. This is well known for the experts in the field. Here are some publications on this topic by the recognized experts.

Quantifying Defects in Graphene via Raman Spectroscopy at Different

Excitation Energies <https://pubs.acs.org/doi/10.1021/nl201432g>

Quantifying ion-induced defects and Raman relaxation length in graphene <https://www.sciencedirect.com/science/article/abs/pii/S0008622310000138>

Unfortunately, using Raman toward GO is currently the widespread mistake, made by researchers not well familiar with the subject. However, as an expert, I cannot support the

publication, where obviously wrong things are stated.

Answer 1:

We understand the importance of inapplicability of Raman spectroscopic analyses for evaluation of the crystallite size of GO. Thank you for an important indication with the detailed explanation together with the appropriate references. We have removed the Raman spectroscopic analyses for the crystallite size of GO from the revised manuscript as you recommended.

Reviewer #2

Comment 1:

The authors have revised the manuscript to my satisfaction. I recommend publication in Nature Communications.

Answer 1:

Thank you very much for recommendation of publication.

Reviewer #3

Comment 1: Homogenize the symbols used in figure 2 so that the symbology remains constant for figures 2,3, 4...etc

Answer 1: We homogenized the symbols used in the figures related to the ripening to avoid misunderstandings as you suggested. Thank you for your suggestion.

Comment 2: Merge supplementary Figure 6 with Figure 5 so that a figure of magnetic and electrical properties is included in the main manuscript.

Answer 2: Thank you for your suggestion. We merged the electrical resistivity data with those in Fig. 5 in the main manuscript and added short explanations as follows.

lines 227-231

future. Here, we extrapolated the plot of electrical resistivity vs. reciprocal compression pressure P_c to the ordinate ($P_c = \infty$) to obtain the minimum contact resistance value, although significantly does not markedly depend on P_c . Fig. 5f shows the extrapolated resistivity. The resistivity at $P_c = \infty$

gradually decreases with ripening and drops for rGO, indicating a significant difference between three the GOs and rGO.

Fig. 5 | Magnetic and electrical properties of freeze-dried GO ripened at 348 K and rGO. a, c $M-H$ curves of the freeze-dried GO ripened at 348 K and rGO at 298 K and 1.8 K, respectively. **b** Changes in the magnetization at 70000 Oe. **d** Changes in the magnetic susceptibility of the GO ripened at 348

K from +500 Oe to −500 Oe. Measuring temperature: ●, 298 K and ○, 1.8 K. Magnetizations and magnetic susceptibilities of the rGO at 298 K and 1.8 K are designated as ■ and □, respectively. e Electrical resistivities of pelletized GO samples as a function of the reciprocal of compression pressure (P_c). The electrical resistivities were measured at compression pressures in the range of 10–240 MPa via the two-electrode method. f Changes in the extrapolated electrical resistivity of the freeze-dried GO ripened at 348 K and rGO. Here, the iGO sample is the non-ripened GO and the mGO and tGO samples are the GOs ripened at 348 K for 12 h and 168 h, respectively.

Comment 3: Migrate Supplementary Table 2 to the main manuscript. It is my opinion that this table summarizes important characteristics of the studied materials and that important conclusions are drawn from it, therefore its inclusion in the main text is important. Moreover, include a column of rGO in this table to compare it with iGO, mGO and tGO.

Answer 3: Thank you for your recommendation. We transferred the Supplementary Table 2 to the main manuscript as Table 1. We have also added the data for rGO in the revised Table 1.

Table 1 | List of the important properties of GOs.

GO	iGO	mGO	tGO	rGO [‡]	Sigma	NS
$\pi - \pi^*$ (nm) (UV-Vis)	230.5 ± 0.5	< 230.0	> 228.8 [†]	N/A	229.6 ± 0.2	229.6 ± 0.2
Reliability of peak deconvolution	High	Medium	Low	N/A	Medium	Medium
Oxygen% (XPS)	31 ± 1	< 30		18	27	27
Epoxy% (XPS)	33 ± 4	23 ~ 29	< 23	12	24	29
Hydroxy% (XPS)	7 ± 4	13 ~ 30	30 ± 2	24	27	6
χ (10 ⁻⁸ emu g ⁻¹) (SQUID)	0.5 ~ 1.8	0 ~ 2.3	0.4 ~ 0.7	0.4	0.83	21.9
Average number of stacking layers (XRD & TEM)	13 ~ 14	8 ~ 13	6 ~ 8	N/A [§]	12	8
Electrical resistivity (Ω m)	> 0.4	0.1 ~ 0.4	< 0.1	2 × 10 ⁻⁶	0.31	2.6

[†]The peak position of the $\pi - \pi^*$ transition of the tGO region overlaps with those of iGO and mGO.

[‡]Data for rGO obtained in this study using rGO, which is reduced iGO at 623 K under an Ar flow.

[§]The XRD pattern of rGO is completely different from those of other GO samples, as shown in Supplementary Fig. 6. The 001 peak of the rGO sample is quite small and the 002 peak is too broad to determine the accurate average number of stacking layers.

REVIEWERS' COMMENTS

Reviewer #1 (Remarks to the Author):

Authors have correctly addressed the question, which was causing the major problem in the two previous submissions. The newly revised version of the manuscript can be published in its present form.